# Nutrient Composition of Autochthonous Beef from Southwest Spain

**DOI:** 10.3390/foods14223961

**Published:** 2025-11-19

**Authors:** Miguel Ángel Cantarero-Aparicio, Manuel García-Infante, Carlos Álvarez, Oliva Polvillo, José Manuel Perea, Alberto Horcada

**Affiliations:** 1Departamento de Producción Animal, Universidad de Córdoba, 14071 Córdoba, Spain; t42caapm@uco.es (M.Á.C.-A.); jmperea@uco.es (J.M.P.); 2Department of Agronomy, School of Agricultural Engineering, University of Seville, Ctra. Utrera km 1, 41013 Seville, Spain; mgarcia53@us.es; 3Department of Food Quality and Sensory Science, Teagasc Food Research Centre, Ashtown, D15 DY05 Dublin, Ireland; carlos.alvarez@teagasc.ie; 4Centro de Investigación, Tecnología e Innovación, Universidad de Sevilla, Avda. Reina Mercedes 4-B, 41012 Sevilla, Spain; oppolo@us.es

**Keywords:** Retinta, Pajuna, Marismeña, Berrenda en Colorado, Lidia, native breed, mineral fatty acids, volatile compounds, sustainable livestock

## Abstract

The aim of this study was to evaluate the nutritional composition of beef from five autochthonous calving breeds from Southwest Spain (Retinta, Pajuna, Marismeña, Berrenda en Colorado, and Lidia) reared under their traditional production systems. *Longissimus dorsi* samples were analyzed for pH, fat, moisture, ash protein content, mineral composition, fatty acid profile, and volatile compounds. Carcass weights of calves ranged from 108 to 328 kg according to the Spanish market, with significant breed differences in fat (range 2.98–8.41%), moisture (69.47–72.62%), and protein (20.98–23.82%), but not in ash (1.03–1.17%). Sodium levels were below 120 mg/100 g, allowing all breeds to be classified as low-sodium, while phosphorus values supported a high-phosphorus label. The Pajuna, Berrenda en Colorado, and Lidia breeds showed higher levels of beneficial fatty acids such as EPA, DPA, DHA, and CLA, with n-6/n-3 ratios ≤ 4, while Retinta and Marismeña presented higher ratios (6.09 and 5.23, respectively). The breeds from Southwest Spain stand out for their content in ketone, ester, and aromatic hydrocarbon volatile compounds linked to the intake of grass, forage, and food concentrate. These results highlight the favorable nutrient profiles and distinctive traits of Spanish autochthonous cattle breeds, emphasizing their value in sustainable production and conservation programs.

## 1. Introduction

At the end of the year 2023, consumption per capita of fresh beef in Europe stood at 9.8 kg/person/year [1]. However, in Spain, this value was only 3.7 kg/person, which represents 3.4% less than in 2022, the equivalent of consuming 0.13 kg less per person-year [2]. In all likelihood, the recent decrease in red meat consumption can be related to the implications of fat intake on human health [3]. Several studies [4,5] suggest that providing accurate information on the nutritional value of meat is essential for increasing consumer confidence and for dieticians and health practitioners to create balanced diets for their patients.

From a livestock point of view, Spain is characterized by its breed diversity. In fact, with a total census of 6,294,640 bovines registered in 2023 [6], there are 47 bovine breeds recognized in Spain in 2024, of which 40 are native breeds (although they only represent 7.81% of the bovine total census), and 7 originate from neighboring countries [7]. In Southern Spain, husbandry practices, according to traditional techniques, are followed for beef production in several scenarios. These systems consider a number of factors, such as breed, feeding method, gender, and age of slaughter of calves, which can cause differences in the nutritional beef composition. Among the native breeds, some are considered in danger of extinction because they have a small number of registered individuals in the official breeds catalog. Specifically, in the South of Spain, of the 7 registered native breeds, i.e., Berrenda en Colorado, Berrenda en Negro, Marismeña, Negra Andaluza, Cárdena Andaluza, Pajuna, Retinta, and Lidia, only the last two breeds show a sufficient number of reproducing animals to warrant their survival in the medium term [8]. The Livestock Associations and the Government of Spain have been deploying strategies for the conservation of native breeds. One example is the recent creation of the quality label “100% native breed” to protect the breed purity of Spanish farmed animals [7].

Southwest Spain encompasses a large area where the production system is representative of Mediterranean beef. This region includes the agroforestry system known as dehesa [9] and mountain areas where autochthonous cattle breeds (Retinta, Pajuna, Berrenda en Colorado, and Lidia, among others) are raised for beef production. In addition, in Southwest Spain, important areas of the Guadalquivir Marshes serve as a sanctuary for the conservation of the native breed Marismeña. However, these native breeds are sometimes crossed with highly productive breeds such as Limousin or Charolais to increase production yields.

According to the traditional system of production in this area, calves remain with their mothers until approximately 6–8 months of age. From this moment on, they are weaned and raised using pasture, forage, or concentrate. Usually, the animals are free-range up to three months before slaughter, at which time the calves receive a concentrated feed supplement to improve weight gain and fat cover. Unlike in Northern Europe or America, cattle in Spain are usually slaughtered at an earlier age (around 12–14 months old) to obtain light carcasses, with a correspondingly lower environmental footprint [10]. According to the production performance of each breed, calves are slaughtered at a different age to fit in with the commercial categories (around 300 kg per carcass) that correspond to the consumer demand from Mediterranean areas.

Curiosity to know the composition of meat has developed widely in the world. For example, the quality and nutrient composition of “Tropical beef” from cattle raised in the tropics areas was reported by Rubio et al. [11]. Among the previous studies on the nutrient composition of beef from European breeds [12], in Spain, studies have been carried out on the nutritional value of autochthonous beef on Northern breeds, associated with mountain production systems [13]. However, no studies focused on understanding the nutritional value of the meat of the native bovine breeds in Southwest Spain have yet been carried out. Considering the uniqueness of the breeds, raised in traditional production systems, and their endangered status, considerable attention and research are required to harness their potential benefits for the environment and meet the consumer demand for high-quality beef. Therefore, the aim of this work was to assess the nutritional composition of several autochthonous breeds from Southern Spain, raised following traditional production systems, to update the national food composition databases and to contribute towards preserving the genetic heritage of native Spanish cattle breeds.

## 2. Materials and Methods

### 2.1. Animals

In this study, a total of 29 cattle from five autochthonous breeds from Southern Spain were selected as follows: two breeds in no danger of extinction (Retinta, n = 8; Lidia, n = 8) and three endangered breeds (Pajuna, n = 7; Marismeña, n = 3 and Berrenda en Colorado, n = 3). Selected breeds are categorized as Mediterranean breeds from Southwestern Europe raised under a traditional feeding system using forage and pasture, according to Piedrafita [10]. A small number of samples of Marismeña and Berrenda en Colorado breeds are presented because these are breeds in grave danger of extinction, and it is extremely difficult to obtain samples due to the small number of specimens. Furthermore, since this document aims to report on the characteristics of the usual meat produced in Southwest Spain, the selection of animals was carried out randomly from different farms according to the production system for each breed under study, taking into account the sex and the animals’ typical slaughter age.

The geographical distribution of the breed in this study is presented in Figure 1.

Animals were selected from farms applying the traditional wild cattle production system as follows: Pajuna, Marismeña, Berrenda en Colorado, and Lidia calves were weaned from 7 to 8 months of age and raised outdoors for a whole year of grazing; four months prior to slaughter, the animals were supplemented with forage and concentrates. In the case of Retinta, the animals were weaned from 6 to 7 months of age and kept indoors using forage and concentrates until slaughter day.

Males of the Retinta, Marismeña, and Berrenda en Colorado breeds were selected for this study, while the Pajuna breed was castrated, and females of the Lidia breed were selected because these are the ones intended for meat consumption. The general characteristics of the cattle from each breed and specific cold carcass weight are shown in Table 1.

### 2.2. Slaughter of Animals and Sample Collection

This study was conducted in several calve farms in Southwest Spain. All animal manipulations in this study were approved by the Ethics Committee of the University of Córdoba (Ref. CEIH-24-01). At the slaughterhouse, the animals were stunned using a captive-bolt stunner and killed following the current European regulations on animal welfare [14]. After slaughtering, the carcasses were chilled and stored at 4 °C for approximately 24 h. Afterwards, the carcasses were split along the spinal column into two equal parts. The two half-carcass parts were weighed to record cold carcass weight. After that, the ultimate pH_24h_ was measured in the caudal area of the *Longissimus dorsi* (LD) muscle from the left half of the carcass. To carry out the pH register, a Crison pH meter was used (Crison Instruments, S.A., Barcelona, Spain) with a temperature compensation probe. From the left half carcass, LD muscles were obtained to determine nutritive parameters. A slice of LD muscle around the 12th rib was obtained and vacuum-packed for analysis at −18 °C. Afterwards, samples of LD muscle were ground to be analyzed for proximate composition (moisture, ash, protein, and fat content), mineral and fatty acid profile analyses.

### 2.3. Proximate Composition

Prior to proximate composition analysis, samples from LD muscle were thawed under vacuum conditions for 24 h at 4 °C. After that, the samples were minced using a Veo Home moulin 200 W grinder (Veotech Inc., KWG-130B, Vannes, France). All the analyses were performed in triplicate. The ash content was determined according to AOAC 920.153 [14], using an electric hot plate and muffle furnace (Carbolite ELF 11/14B, Sheffield, UK) at 550 °C for 12 h. The fat content was determined in pre-dried samples using an MQC+ benchtop nuclear magnetic resonance fat analyzer (BRUKER Corporation, Coventry, UK). The samples were weighed pre- and post-drying at 105 °C for 24 h in an Heraeus oven (Thermo electron Corp., Barcelona, Spain) to measure the moisture content [14]. The protein content was analyzed following the AOAC [15] procedure, using a 2300 Kjeltec Analyzer Unit (Foss Tecator, Höganäs, Sweden). A conversion factor of 6.25 (N×6.25) was used for nitrogen to protein content in meat [15]. The chemical composition of the meat was expressed as % of fresh meat.

### 2.4. Mineral Composition

For mineral profile analysis, samples of approximately 1 g of LD muscle were thawed under vacuum conditions for 24 h at 4 °C. Minced samples were mixed with 4 mL of concentrated nitric acid. The vessels were closed tightly and placed in the microwave oven. Hydrolysis was undertaken using a closed-vessel microwave digestion system (Milestone ETHOS 1 Series, Sorisole 24010, Italy) for 2 min at 220 °C. After digestion, the vessels were cooled to room temperature and opened. The digested samples were diluted with deionized water and transferred to analysis tubes. Detection and quantification of the mineral profile were performed using the ICP-OES method in an ICP Spectro blue (Spectro Analytical Instruments GmbH, Kleve 47533, Germany) in the case of calcium, iron, potassium, magnesium, sodium, phosphorus, selenium, and zinc [16]. The mineral content of the meat was expressed as mg/100 g fresh meat.

### 2.5. Fatty Acid Profile

Intramuscular fatty acid methyl esters (FAMEs) were analyzed following the method proposed by Gutiérrez-Peña [17]. Approximately 1 g of LD sample was thawed and saponified in 6 mL of 5 M KOH in methanol/water (50:50 *v*/*v*) with hydroxyquinone (1 g/L) at 60 °C for 1 h, followed by flushing with nitrogen, after which the mixture was diluted with 12 mL of 0.5% NaCl and 5 mL of petroleum ether, and the non-saponifiable fraction was removed. Next, 3 mL of glacial acetic acid was added to neutralize the KOH. Double petroleum ether washing was used for fatty acid isolation. A nitrogen jet was used to evaporate the organic solvent. The fatty acids extracted were methylated using 200 µL of TMS-DM in methanol/toluen (2:1, *v*/*v*) at 40 °C for 10 min, dried under nitrogen, and dissolved in 1 mL of n-hexane containing 50 ppm of butylated hydroxy toluene. The samples were centrifuged at 15,000 rpm for 5 min. Supernatant containing FAMEs was transferred for analysis to 2 mL vials. The separation of FAMEs was carried out using a gas chromatograph Agilent 6890N Network GS System (Agilent, Inc., Santa Clara, CA, USA), equipped with a flame ionization detector (FID) and fitted with a HP-88 capillary column (100 m, 0.25 mm i.d., 0.2 μm film thickness, Agilent Technologies Spain, S.L., Madrid, Spain). An HP 7683 autoinjector was used to inject the samples. The chromatographic conditions were as follows: initial column temperature 10 °C, programmed to increase at a rate of 3 °C/min to 158 °C and then at 1.5 °C/min to 190 °C, maintaining this temperature for 15 min, then at 2 °C/min to 200 °C, and then increasing again at 10 °C/min to a final temperature of 240 °C for 10 min. The temperatures for injection and FAME detection were kept at 300 and 320 °C, respectively. Hydrogen was used as the carrier gas at a flow rate of 2.7 mL/min. The split ratio was 17.7:1, and 1 µL of solution was injected. Nonadecanoic acid methyl ester (C19:0 ME) at 10 mg/mL was used as internal standard. Individual FAMEs were identified by comparing their retention times with those of authenticated standards from Sigma (Sigma Chemical Co., Ltd., Poole, UK). The fatty acid profile of the intramuscular fat was expressed as mg fatty acid/100 mg fresh muscle.

### 2.6. Volatile Compound

The volatile compounds of the meat were identified using the solid phase micro extraction (SPME) analysis technique [18]. To achieve this, a sample of 20 g from the LD muscle was thawed at 4 °C overnight. Next, the sample was cooked at 200 °C under a mixed closed griddle (Jatta electro, GR266 1000W, Abadiano, Vizcaya, Spain) for three minutes to a sample core temperature of 70 °C. Quickly, the meat was chopped finely and, together with all the fat released from the steak during cooking, placed in an electric bowl chopper (Janke and Kunkel A-10, IKA Labortechnik, Staufen im Breisgau, Germany). Straight after chopping, 10 g of the cooked sample was placed in a headspace vial (Tekmar, 20 mL) and equilibrated for 40 min at 40 °C prior to exposure of the SPME fiber (Fiber Assembly 50/30 μm DVB/CAR/PDMS, Stableflex -2 cm- 23 Ga, Gray-Notched; Bellefonte, PA, USA) placed over the sample for a further 20 min. The DVB/CAR/PDMS fiber has been reported as one of the most efficient coatings for the extraction of volatile compounds from cooked beef cuts, providing good reproducibility and a broad extraction range. Moreover, optimization studies recommend fiber coatings and extraction times in the range of 30–40 min for cooked meat matrices at 60 °C [19]. The analysis of the volatile compounds was performed using a Thermo Scientific TRACE 1300 series (Milan, Italy) gas chromatograph (GC) equipped with a Thermo Scientific TRIPLUS RSH autosampler (Milano, Italy) for injection, and coupled with an ion trap mass spectrometer (Thermo Scientific ISQ QD Single Quadrupole Mass Spectrometer; Milan, Italy). The desorption process of the compounds was carried out using the splitless mode with purges of 5 mL/min. Injection was maintained for 10 min at a temperature of 40 °C. The chromatograph oven was heated to 40 °C and, once the sample had been released into the gas phase, the temperature was increased at a rate of 5 °C/min up to a maximum temperature of 220 °C. As the carrier gas, helium at 20 psi with a flow of 1.2 mL/min. at 40 °C was used. A VF-WAXms fused silica capillary column (30 m length × 0.25 mm id × 0.50 μm film thickness, Agilent Technologies, Inc., 2012, Santa Clara, CA, USA) was used to separate the volatile compounds. The operating conditions were as follows: initial temperature, 40 °C for 5 min., then increased to 220 °C at a rate of 5 °C/min. and held for 5 min, with a total acquisition time of 56 min. The detection conditions were as follows: the temperatures of the source and quadrupole were 175 °C and 150 °C, respectively. The mass spectra of the volatile compounds were generated by an MS equipped with an ion trap. The data acquisition was performed by scanning the mass range 29–400 amu. in EI mode (70 eV with an emission current of 50 mA) at 1.9 microscans/s. The volatile compounds were identified by comparing their mass spectra with spectra included in NIST/EPA/NIH Mass Spectral Libraries and confirmed by matching their linear retention indices (LRI) [20] or the Flavornet database. The linear retention indices (LRI) [21] were calculated by previous injection of standards of saturated n-alkanes (C6-C22) under the same GC–MS conditions. The volatile compounds were expressed as mean peak area × 10^3^.

### 2.7. Statistical Analysis

Data were analyzed using one-way ANOVA followed by a post hoc Tukey test (*p* < 0.05) to compare differences among breeds. Before performing the ANOVA and PCA, data were tested for normality and homogeneity of variances using the Shapiro–Wilk and Levene’s tests, respectively. Outliers were examined using the interquartile range (IQR) method, defining potential outliers as values outside 1.5 × IQR from the first or third quartile. No data were excluded, as all observations were within biologically plausible ranges. Data were normalized to have 0 mean and 1 standard deviation. Principal component analysis (PCA) was performed, and principal components with Eigenvalues greater than 1 were selected. The 24 chosen components accounted for 98.91% of the accumulated variance. Discriminant analysis (DA) was conducted using the functions of the principal components, labeled according to the breeds studied. The stepwise forward method was applied, with selection criteria based on an F value to enter of 5.5 and an F value to remove of 4.0. The analyses were performed using the Statgraphics Centurion 18 statistical package [22].

## 3. Results

### 3.1. Discriminant Analysis

The results of the discriminant analysis, including (as variables) carcass weight, pH, proximal composition, mineral composition, and fatty acid profile, are shown graphically in Figure 2. Such analysis allows the cattle breeds to be placed in specific regions of the plot; in this case, four breeds are grouped individually in each quadrant of the plot, while the Pajuna breed is located between two quadrants; nevertheless, a clear separation is observed with no overlapping.

Function 1 (vertical axis) explains 55.8% of the variability among breeds, differentiating the breeds according to carcass weight, protein and ash content, several mineral concentrations (particularly K, Na, Zn, and P), and several polyunsaturated fatty acids (PUFAs), including C18:3n-6, C20:3n-3, and C18:2n-6. This horizontal separation clearly discriminated between breeds not at risk of extinction (Retinta and Lidia), located on the right side of the plot, and endangered breeds (Berrenda en Colorado, Pajuna, and Marismeña), positioned on the left. Function 2 (horizontal axis) explains 25.2% of the variability among the studied breeds. This function was primarily associated with protein and fat content in meat, carcass weight, only one mineral concentration (calcium), saturated fatty acid (C20:0), monounsaturated fatty acid (C14:1), and several polyunsaturated fatty acids (C18:2n-9, C20:3n-3, C20:3n-6, and C20:5n-3 [EPA]). Along this axis, temperamental and wild breeds (Lidia and Marismeña) were located towards the lower part of the plot, while docile breeds selected for meat production (Retinta and Berrenda en Colorado) appeared in the upper region. The Pajuna breed occupied an intermediate position, reflecting its non-temperamental nature, as these animals are typically castrated and used as working animals. Overall, the results suggest that both temperament and conservation status influence carcass and meat composition, but through different mechanisms: temperament via physiological and metabolic reactivity, and conservation status through selection history, management system, and ecological adaptation, rather than productivity alone. It is remarkable that function 1 and function 2 include carcass weight, protein, and several polyunsaturated fatty acids in meat to explain variability among calf breeds from Southwest Spain.

It should be noted that Retinta and Lidia breeds are good performers in terms of production yields, and improvement plans are in place to increase the yields even further [10]. However, in Berrenda en Colorado, Pajuna and Marismeña, any improvements in production yield are limited because of their low profitability, which reduces the interest in increasing their numbers in the farm system.

### 3.2. Carcass Weight and Proximate Composition

The range of carcass cold weight ranged from 108 to 328 kg (Table 2), with the Marismeña breed exhibiting the highest carcass weight (327.67 kg), followed by Retinta (301.40 kg) and Pajuna (276.49 kg), while the Lidia breed showed the lowest mean weight (108.22 kg). This corresponds to the fact that Marismeña, Retinta, and Pajuna breeds are farmed for meat production, while the Lidia breed is intended for bullfighting festivals, in which smaller animals are used [23]. While meat breeds exhibit rapid growth, the breed used for bullfighting generally grows slowly, resulting in a longer production cycle that generates greater muscle development and reduced fat content [24].

The pH and proximate composition of meat from the five Spanish autochthonous breeds in this study, reared under their respective traditional production systems, are presented in Table 2. The pH values, registered 24 h post-mortem, were found to be within the physiological range considered normal for fresh meat [25], which is indicative of an appropriate progression of the muscle to meat conversion process. The pH values are within the range described as normal values for fresh meat (5.5 to 6.0) from animals that have not been stressed prior to slaughter [25]. Therefore, no alterations in the meat are expected that could hinder the study of the meat characteristics.

Significant differences (*p* < 0.001) were observed in intramuscular fat, moisture, and protein content (Table 2), whereas ash content did not show statistically relevant differences (*p* > 0.05). Ash content exhibited limited and homogeneous variation across the groups (1.03–1.17%).

The fat content of the meat (intramuscular fat, IMF) showed a high variability among breeds (*p* < 0.001; Table 2). The values recorded in this study are consistent with previously reported ranges in Spanish and other European breeds, which span from 1.12 to 9.2 g/100 g fresh meat [13,26]. The Pajuna breed exhibited the highest IMF content (8.41%), comparable to the mean values reported in crossbred or purebred selected British Hereford cattle (7.6%), although lower than those observed in geographically adjacent breeds such as Portuguese Alentejana cattle (9.76–13.03%) finished under different feedlot systems. The values for fat content in meat recorded in Pajuna could be attributed to the combined effect of the calves’ castration, prolonged concentrate feeding, and an intermediate slaughter age. An extensive amount of the literature describes that castration of young male calves has been found to enhance the accumulation of intramuscular fat in beef because castration technique decreases lipolysis in animal tissues, while lipogenesis and lipid absorption increase [27]. The Berrenda en Colorado breed showed elevated IMF (5.56%), possibly associated with an older slaughter age (over 3 years). Meat from Marismeña and Retinta exhibited intermediate values for IMF content among the five breeds studied (4.62% and 3.90%, respectively), with both breeds presenting results higher than those reported by Campo et al. [13] for autochthonous bovine breeds such as Pirenaica (Spain) or Bruna d’Andorra (Andorra). Although the Lidia-breed animals were females, this breed showed the lowest IMF content (2.98%), a result consistent with an extensive production system characterized by high grazing utilization and low energy supplementation. In line with the above, the results observed in the Lidia breed were also higher than those reported by Nogales et al. [28], who reported an IMF of 1.22%. Similarly, Humada et al. [29] showed lower fat content values in meat from calves from Rubia Gallega (0.86 to 1.15%) and Tudanca (ranging from 1.14 to 1.31%) raised in pasture in Northern Spain. It is noteworthy that the Lidia breed was the only breed evaluated in which females were included; however, the low energy availability could attenuate the physiological effect of sex on lipid deposition [30]. According to European regulations [31], only Lidia meat could be classified as “low fat” (<3 g/100 g fresh meat).

Previous studies have shown that carcasses from low-fat animals have higher moisture values [32]. Moisture values in the meat from the five breeds used in this study exceed those reported by Horcada et al. [33] for French Charolais and Limousin breeds (selected for meat production) that were raised in the same region as the present study (Andalusia, Spain). Moisture content in the meat was highest in the Lidia breed (72.62%) and lowest in Pajuna (69.19%) (Table 2). The results found could be consistent with the inverse relationship between moisture and fat described by Nian et al. [27]. Nevertheless, these values were lower than those reported by Brugiapaglia et al. [34], who indicated moisture contents ranging from 74.6 to 75.38% in meat from Italian and Friesian breeds; as did Panea et al. [35], who reported a range between 73.57 and 75.96% in calves from autochthonous Retinta Spanish breed, as well as various commercial crossbreeds as Limousin × Retinta, Pirenaica × Retinta, or Asturiana de los Valles × Retinta.

The overall protein content in meat (21–24%) was consistent with previously reported ranges for Spanish breeds, including Rubia Gallega, Retinta, and various crossbreeds [13,36]. The average protein content in the Pajuna breed was significantly lower (20.98%) compared to the other Northern Spain breeds, such as Pirenaica (ranging from 23.2 to 24.1%), as described by Campo et al. [13], and the other four breeds analyzed here. Several factors may contribute to this result, where the lower protein content shown by Pajuna (20.98%) could be due to a higher intramuscular fat content, an inverse relationship which has been previously reported [37]. The Lidia breed, despite its advanced age, showed no significant protein content (22.61%) compared to the other breeds, which is consistent with the lean character of this breed and the production system, mainly based on grazing [38].

In general, all the breeds analyzed here provide a consistent amount of protein, with very little difference in its composition. Fat content was the most notable difference, since castrated animals were sampled and are prone to fat deposition [27]. The second most noteworthy difference was the final carcass weight, which is a key factor when it comes to production performance and designing improvement programs for beef production.

### 3.3. Mineral Composition of Meat

In recent years, there has been sustained interest in generating data on trace elements in meat to update national databases, highlighting the need for updated information on the mineral composition of different breeds across traditional production systems [13,39]. The mineral content in the green diet of ruminants can affect the final composition of the meat [40]. On the other hand, Reiné et al. [41] reported that mineral content in plants is highly variable and depends on factors such as the plant species and the ecological conditions under which they grow. Consequently, as observed in other breeds raised on green pastures in northern Spain, a relationship between plant mineral composition and beef mineral content could also be expected in local calf breeds from Southwestern Spain.

The results regarding mineral composition of autochthonous breeds in Southwest Spanish breeds are shown in Table 3. Overall, mineral levels were comparable to, or higher than, those reported by Campo et al. [13] from Pyrenean breeds located in the North of Spain (Pirenaica, Bruna d’Andorra, and Gascon oxen). Calcium and sodium were the only evaluated minerals that showed statistically significant differences among Southwest Spanish breeds (*p* < 0.05).

In order to guarantee health of consumers, the European Food Safety Authority (EFSA) [42] reported the dietary reference intakes level of trace mineral in meat as following: daily Ca intake < 2500 mg/d and Na < 2000 mg/d; P around 550 mg/d; K around 3500 mg/d; Mg around 250 mg/d; Zn < 25 mg/d; Se around 255 µg/d and Fe between 6 and 13 mg/d. Regarding calcium content, the Retinta breed presented the highest level (11.58 mg/100 g), in contrast with Berrenda en Colorado, which exhibited the lowest value (4.83 mg/100 g) (*p* < 0.05). These results fall within the range reported by Campo et al. [13] for different breeds raised in Spain, such as young Pirenaica bulls (4.29 mg/100 g), Bruna d’Andorra (7.72 mg/100 g), and Gascon oxen (7.72 mg/100 g). Calcium is a mineral whose concentration may vary with the animal’s age [43]. However, evidence of a relationship between the age of the animal and calcium content in the beef from Southwest Spanish breeds was not present. Since the calves were raised in several ecological areas (dehesa, mountain, and marshes), it is likely that the differences detected in calcium content in beef could be attributed to heterogeneity in pastures where animals were raised and feeding regimes [44]. However, meat is not the main source of calcium in the diet, and these differences have little impact on its nutritional value.

According to regulations on nutrition claims established by the European Commission [31], intake of “low in sodium” (<120 mg/100 g) content in meat to prevent cardiovascular health diseases is proposed. The sodium content in the meat varied significantly among breeds (*p* < 0.01), with values ranging between 52.31 mg/100 g (Lidia) and 72.06 mg/100 g (Marismeña). These data are consistent with those reported by Campo et al. [13] in Bruna d’Andorra (56.1 mg/100 g) and Gascon oxen (54.9 mg/100 g). The importance of environmental factors, such as the mineral composition of pastures, is well established and could explain the observed differences in sodium content in the beef from Southwest Spanish autochthonous breeds [41]. The high sodium concentration recorded in meat from the Marismeña breed may be related to the particularly sandy ecosystems near the sea where this breed is typically reared. Phosphorus concentrations in beef from Southwest Spanish autochthonous breeds, which ranged between 220 and 235 mg/100 g, mean that the product is considered a “high phosphorus content” food [31].

The iron content ranged from 1.64 to 2.71 mg/100 g, which aligns with the values reported by Purchas et al. [45] in New Zealand steers (1.91 mg/100 g) and is similar to those described by Campo et al. [13] in Pyrenean breeds (1.39–2.68 mg/100 g). Although it has been suggested that iron content increases with the age of the animal [43], no such trend was observed in this study. In fact, no significant differences in iron content in beef (*p* > 0.05) were observed among the different breeds studied, despite the animals being slaughtered at different ages, in accordance with the traditional breeding system of the different breeds, although on average, the Lidia breed (with the highest slaughter age and the leanest meat) showed a trend to have the highest iron content. This could be attributable to relative similarities in grazing systems and the level of rusticity among the breeds evaluated [46]. However, the higher iron content observed in Lidia meat can be attributed to a combination of physiological and management factors. Lidia cattle are raised under extensive conditions and are subjected to high physical activity, which promotes oxidative muscle metabolism and increases myoglobin concentration, a key source of heme iron. Their meat is also leaner, resulting in a greater proportion of iron-rich muscle tissue relative to fat. In addition, the pronounced stress response typical of this breed may lead to incomplete exsanguination during slaughter, contributing to higher residual heme and non-heme iron. Together, these factors account for the elevated iron levels and darker meat color characteristic of this temperamental and physically active breed.

The magnesium levels in beef from autochthonous Southwest Spanish breeds analyzed were within the range described by Campo et al. [13] for autochthonous Northern Spanish breeds (20.80 to 26.5 mg/100 g edible meat). Magnesium levels have not been affected by the breed, probably because all calves ingested forage, and pasture was included in the diet. The zinc content in beef from Southwest Spanish autochthonous breeds ranged from 4.8 to 6.7 mg/100 g meat, also consistent with the findings reported by Campo et al. [13] for breeds from Northern Spain. These results confirm that these breeds are a good source of Zn, of which the daily requirement is approximately 14 mg/day and 8 mg/day for adult men and women, respectively [47].

### 3.4. Fatty Acid Composition of Meat

The content of fatty acid per gram of edible portion of meat (muscle + visible fat) from autochthonous Southwest Spanish breeds is shown in Table 4. The beef obtained from Southwest Spanish breeds had a lower amount of total FAMEs per gram of fresh meat (ranging from 5.75 to 12.28 mg/g fresh meat) than reported by other autochthonous breeds from Northern Spain, as reported by Moreno et al. [36] in calves from the breeds Rubia Gallega (around 21.23 mg/g fresh meat), Pirenaica (around 22.27 mg/g fresh meat) [13], Asturiana de los Valles or Asturiana de la Montaña (ranging 12.02 to 25.20 mg/g fresh meat) [12].

Among the breeds analyzed in this study, the Marismeña breed displayed the lowest total content of FAMEs (total SFA, MUFA, and PUFA; 5.75 mg/g fresh edible meat) per gram of edible meat, whereas Berrenda en Colorado showed the highest SFA and MUFA content (6.38 and 5.12 mg/g of fresh meat, respectively).

Interestingly, meat from the Lidia breed had the lowest total fat content (2.98%; Table 2), yet it exhibited significantly higher levels of SFA, MUFA, PUFA, and total FAMEs compared with the other autochthonous breeds from southern Spain, second only to the Barrenda en Colorado breed (Table 4). These results suggest that the FAME content in the edible fat of the Lidia breed is higher than that of other local breeds, whereas other lipids, such as cholesterol, are lower. The total fat in meat includes all lipids, while FAMEs measure only esterified fatty acids, so fat content and FAME levels can be at variance depending on the lipid composition of the fat. Therefore, in leaner muscles such as those of the Lidia breed, which have a higher relative proportion of membrane phospholipids per unit tissue, compared to intramuscular fat, FAME yields can be comparatively higher even when the total NMR-detectable fat is lower. Conversely, fattier breeds accumulate larger triacylglycerol depots and higher proportions of non-esterifiable lipids (e.g., free cholesterol and sphingolipids), diluting the phospholipid fraction and resulting in lower FAMEs per 100 g tissue [48].

As was reported by Hino et al. [49] and Givens and Gibbs [50], red meat is the main dietary source of 20:5 n-3 (EPA), which has comparable health benefits to those of 22:5 n-3 (DPA) and 22:6 n-3 (DHA) fatty acids and CLA [51]. In general, among the Southwest Spanish breeds studied, the highest content of desirable fatty acids (EPA, DPA, DHA, and CLA) beneficial for human health was observed in the meat from Lidia, Pajuna, and Berrenda en Colorado (Table 4), while meat from the Retinta and Marismeña breeds showed a significantly lower content of desirable fatty acids. The lower levels of EPA, DPA, DHA, and CLA observed in meat from the Retinta breed, compared with the other beef breeds analyzed, are likely related to differences in forage and concentrate intake, whereas a diet based on fresh pasture is associated with higher concentrations of these beneficial fatty acids in meat.

In order to prevent cardiovascular diseases in humans, several studies have suggested that a maximum n-6/n-3 PUFA ratio of 4 is recommended for human health benefits [52]. Meat from the Pajuna, Berrenda en Colorado, and Lidia breeds exhibited a more favorable n-6/n-3 ratio than Retinta and Marismeña breeds. Specifically, the first three breeds showed ratios below 4, considered beneficial, whereas Retinta and Marismeña had higher ratios ranging from 5.23 to 6.09 (Table 4). Overall, Southwest Spanish breeds displayed low and health-promoting n-6/n-3 PUFA ratios, likely reflecting their traditional diet of fresh pasture or forage supplemented with concentrates.

### 3.5. Volatile Compounds Profile

Following the analysis of proximate composition, lipid profile, and mineral content, a profile was conducted for volatile compounds, which are closely related to nutritional quality, fatty acid preservation, and the overall health value of beef, as well as their origin and production system.

A total of 163 volatile compounds (Appendix A) were identified and quantified across the five autochthonous breeds studied, and grouped into 12 functional families (Table 5). These findings make a relevant contribution to the sensory characterization and valorization efforts of endangered autochthonous cattle breeds from Southwest Europe [53,54]. The volatile profile observed represents a complex mixture of carboxylic acids, alcohols, aldehydes, ketones, esters, lactones, sulfurs and nitrogen-containing compounds, pyrazines, pyrroles, furans, and aromatic hydrocarbons. Both the total abundance and the relative contribution of each family exhibited high intragroup variance, resulting in no significant differences (*p* > 0.05) for most of the volatile compound families analyzed, including the major ones (aldehydes, alcohols, and carboxylic acids). This degree of individual heterogeneity has also been observed in studies of other endangered cattle breeds, which, unlike industrial breeds with highly standardized genetics and management, exhibit a greater variability in the sensory properties of cooked meat, particularly in the volatile compounds responsible for meat aroma [55,56]. Such variability is likely linked to the broader genetic diversity and less intensive selection pressures typically found in autochthonous populations of breeds of calves from Southwest Spain.

In contrast to the major volatile compounds identified, the ketone, ester, and aromatic hydrocarbon families displayed significant differences among the autochthonous cattle breeds from Southwest Spain studied (*p* < 0.05). These compounds, although present at lower concentrations within the volatile mixture due to their low odor thresholds, have a sensory impact that is readily identifiable and can be perceived by consumers. Such sensory-active volatiles are capable of releasing distinctive aromatic nuances that help to define breed-specific flavor profiles, which can be perceived and valued by consumers [57,58]. This highlights the importance of minor compounds in differentiating the sensory identity of meat in endangered autochthonous cattle breeds from Southwest Spain, and underscores their potential for product valorization.

These findings suggest that, while the basal meat flavor of autochthonous breeds from Southwest Spain is not markedly influenced by breed, there is a discernible trend in the distribution of the major volatile compound families. Specifically, alcohols represent the predominant family in the Pajuna, Marismeña, Berrenda en Colorado, and Lidia breeds, whereas aldehydes are the most abundant in the Retinta breed. Carboxylic acids were the third most abundant family in the volatile profile of all breeds, except for the Lidia breed, where Carboxylic acids were ranked second in abundance. These compositional differences may contribute to slightly differentiate aromatic profiles among breeds. Furthermore, the subtle differences in aroma are likely driven by minor volatile compounds with low odor thresholds, which play a key role in defining each of the breed-specific aromatic nuances.

The most abundant volatile compounds identified (Appendix A) were ethanol, 1-octen-3-ol, and 2.3-butanediol in the alcohol family; hexanal, nonanal, and benzaldehyde in the aldehyde family; and acetic acid, dodecanoic acid, and tetradecanoic acid in the carboxylic acid family. Studies in Mediterranean cattle breeds, such as Pirenaica and Friesian [59], also identified ethanol as one of the principal compounds in the volatile profile. The aroma of ethanol has been described as sweet, although it is generally considered sensorially neutral at high concentrations. In contrast, hexanal is a well-established indicator of lipid oxidation and is responsible for lending green and fresh aroma notes to meat [60]. Overall, the predominant volatile compounds observed in the meat of the autochthonous Southwest Spanish breeds studied tend to contribute subtle or low-intensity base notes, which is characteristic of Mediterranean cattle breeds raised using pasture and forage in the diet, and can be appreciated by consumers.

## 4. Conclusions

The important concentration of native bovine breeds in the south of Spain is a part of the national heritage. In order to update the national food composition database in Spain, nutritional analysis of beef from five autochthonous breeds located in Southwest Spain was performed. The FAME composition is representative of animals which have been reared on cereal-based diets, forage, and grass. According to nutritional claims, meat from autochthonous Southwest Spanish breeds can be considered as an excellent source of quality nutrients such as iron and zinc, while sodium and selenium contents are low. The wide range of autochthonous breeds from Southwestern Spain provides a response to the variety of consumers’ preferences. Taking all these data into account, the breeds from Southwest Spain stand out for their unique flavors associated with the intake of grass, forage, and concentrate in the diet and a low fat content. Rustic breeds from Southwestern Spain show a highly desirable fatty acid content for human health and a favorable n-6PUFA/n-3PUFA ratio, which helps prevent coronary diseases and could be recommended in clinic diets.

Information on the nutritional value of beef from these native breeds could be included in the future in the labeling of the beef in order to meet consumer demands. Moreover, the knowledge of nutritional traits of beef from autochthonous breeds could be used to improve the conservation programs in these endangered breeds. However, more studies about the nutrient composition of some autochthonous beef from Southwest Spain are required because the information available is dramatically scarce, and traditional production systems are heterogeneous and highly specific for each breed analyzed.

In the near future, it will be important to assess how local feeding systems influence the nutritional value of beef from Southwestern Spain, since changes in husbandry practices driven by new regulations or climate variability could alter the composition of traditional pastures and, consequently, the nutritional characteristics of the meat obtained from the autochthonous breeds in this region.

## Figures and Tables

**Figure 1 foods-14-03961-f001:**
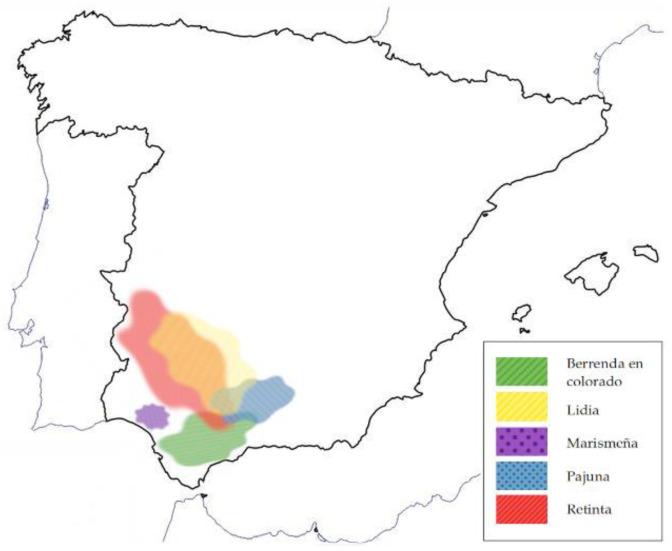
Geographical distribution of the breed system in this study.

**Figure 2 foods-14-03961-f002:**
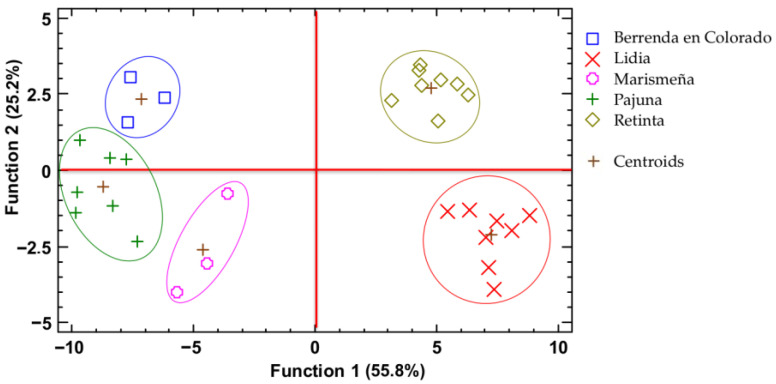
Discriminant analysis plot for the five breeds studied. Color marks represent the values of two discriminant functions for each observation.

**Table 1 foods-14-03961-t001:** General characteristics of the autochthonous cattle produced for each breed and average cold carcass weight of the animal in the study.

	Retinta	Pajuna	Marismeña	Berrenda en Colorado	Lidia
n	8	7	3	3	8
Sex	Male	Castrated	Male	Male	Female
System of feeding	F + C	P + C	P + C	P + C	P + C
Age at slaughter (days)	546 ± 165	722 ± 157	892 ± 27	1201 ± 396	946 ± 204
Cold carcass weight (kg)	301.40 ± 37.29	276.49 ± 30.64	327.67 ± 19.22	235.40 ± 39.30	108.22 ± 19.10

C: concentrate; F: forage; P: pasture.

**Table 2 foods-14-03961-t002:** Carcass weight, pH, and proximate composition (expressed as % of fresh meat) of five autochthonous cattle breeds from Southwest Spain.

	Breed					SE	*p*-Value
	Retinta	Pajuna	Marismeña	Berrenda en Colorado	Lidia		
Cold carcass weight (kg)	301.40 ^a^	276.49 ^ab^	327.67 ^a^	235.40 ^b^	108.22 ^c^	16.58	≤0.001
pH_24h_	5.56	5.66	5.75	5.65	5.71	0.03	0.676
Fat (%)	3.90 ^bc^	8.41 ^a^	4.62 ^bc^	5.56 ^b^	2.98 ^c^	0.44	≤0.001
Moisture (%)	71.20 ^ab^	69.19 ^c^	70.55 ^bc^	69.47 ^bc^	72.62 ^a^	0.30	≤0.001
Protein (%)	23.51 ^a^	20.98 ^b^	23.70 ^a^	23.82 ^a^	22.61 ^a^	0.24	≤0.001
Ash (%)	1.05	1.09	1.06	1.03	1.17	0.02	0.433

SE: Standard error. Different superscripts (^a–c^) indicate significant differences (*p* < 0.05) among breeds.

**Table 3 foods-14-03961-t003:** Mineral composition (expressed as mg/100 g fresh meat) in the *Longissimus dorsi* muscle of five autochthonous cattle breeds from Southwest Spain.

	Breed					SE	*p*-Value
	Retinta	Pajuna	Marismeña	Berrenda en Colorado	Lidia		
Calcium	11.58 ^a^	8.87 ^ab^	8.66 ^ab^	4.83 ^b^	7.90 ^ab^	0.65	0.035
Sodium	57.41 ^ab^	63.26 ^ab^	72.06 ^a^	53.72 ^ab^	52.31 ^b^	1.82	0.012
Phosphorus	235.15	220.67	234.17	222.19	235.47	3.02	0.311
Iron	1.81	2.17	2.55	1.64	2.71	0.14	0.083
Potassium	278.13	263.55	286.18	273.75	265.00	4.14	0.485
Magnesium	24.64	21.97	24.12	23.76	23.15	0.34	0.057
Zinc	5.01	6.03	6.02	4.79	6.76	0.24	0.059
Selenium	traces	traces	traces	traces	traces	-	-

SE: Standard error. Different superscripts (^a,b^) indicate significant differences (*p* < 0.05) among breeds.

**Table 4 foods-14-03961-t004:** Fatty acid profile (expressed as mg of fatty acid per gram of edible portion of meat, inc. muscle + visible fat) in the *Longissimus dorsi* muscle of five autochthonous cattle breeds from Southwest Spain.

	Breed					SE	*p*-Values
	Retinta	Pajuna	Marismeña	Berrenda en Colorado	Lidia		
Total SFA	4.243 ^b,c^	5.451 ^b^	3.021 ^c^	6.379 ^a^	5.620 ^b^	0.27	0.005
C8:0	0.037 ^b,c^	0.050 ^a,b^	0.048 ^a,b^	0.060 ^a^	0.047 ^a,b^	0.00	0.016
C10:0	0.054	0.060	0.056	0.075	0.057	0.00	0.212
C11:0	0.008 ^b^	0.011 ^a,b^	0.016 ^a^	0.010 ^a,b^	0.007 ^b^	0.00	0.003
C12:0	0.065 ^b^	0.069 ^a,b^	0.061 ^b^	0.104 ^a^	0.065 ^b^	0.00	0.005
C13:0	0.013 ^b^	0.019 ^a^	0.011 ^b^	0.013 ^a,b^	0.016 ^a,b^	0.00	0.011
C14:0	0.236 ^a,b^	0.294 ^a,b^	0.140 ^c^	0.359 ^a^	0.206 ^b,c^	0.02	0.005
C15:0	0.051	0.050	0.047	0.065	0.067	0.00	0.117
C16:0	2.084 ^b^	2.735 ^a,b^	1.424 ^b^	3.460 ^a^	2.518 ^a,b^	0.15	0.006
C17:0	0.075 ^a,b^	0.100 ^a,b^	0.065 ^b^	0.117 ^a,b^	0.106 ^a^	0.01	0.010
C18:0	1.553 ^b^	1.922 ^a,b^	1.089 ^b^	2.002 ^a,b^	2.430 ^a^	0.11	0.001
C20:0	0.024 ^b^	0.062 ^a^	0.016 ^b^	0.038 ^a,b^	0.052 ^a^	0.00	0.000
C21:0	0.008 ^c^	0.019 ^b^	0.011 ^b,c^	0.030 ^a^	0.015 ^b^	0.00	0.000
C22:0	0.010 ^b,c^	0.012 ^a,b^	0.012 ^a,b,c^	0.016 ^a^	0.008 ^c^	0.00	0.000
C23:0	0.013 ^b^	0.035 ^a^	0.019 ^a,b^	0.020 ^a,b^	0.010 ^b^	0.00	0.001
C24:0	0.004 ^b,c^	0.006 ^a,b^	0.003	0.009 ^a^	0.014 ^a^	0.00	0.000
Total MUFA	2.862 ^c^	4.480 ^b^	2.162 ^c^	5.124 ^a^	4.220 ^b^	0.25	0.002
C14:1	0.034 ^b^	0.025 ^b^	0.020 ^b^	0.069 ^a^	0.035 ^b^	0.00	0.000
C15:1	0.023	0.025 ^b^	0.031	0.032	0.028	0.00	0.658
C16:1	0.201 ^a,b^	0.338 ^a^	0.151 ^b^	0.409 ^a^	0.217 ^b^	0.02	0.000
C17:1	0.047 ^c^	0.083 ^a,b^	0.050 ^b,c^	0.113 ^a^	0.073 ^b,c^	0.01	0.000
C18:1n-9t	0.080 ^b^	0.185 ^a^	0.044 ^b^	0.096 ^b^	0.082 ^b^	0.01	0.000
C18:1n-11t	0.194 ^a,b^	0.190 ^a,b^	0.138 ^b^	0.251 ^a,b^	0.253 ^a^	0.01	0.033
C18:1n-9c	2.240 ^b,c^	3.577 ^a^	1.702 ^c^	4.107 ^a^	3.479 ^a,b^	0.21	0.002
C20:1n-9	0.025	0.036	0.016	0.017	0.030	0.00	0.075
C22:1n-9	0.016 ^a,b^	0.016 ^a,b^	0.013 ^b^	0.021 ^a,b^	0.021 ^a^	0.00	0.011
C24:1	0.003 ^b^	0.004 ^b^	0.002 ^a,b^	0.007 ^a,b^	0.018 ^a^	0.00	0.014
Total PUFA	0.712 ^b^	0.913 ^a,b^	0.562 ^c^	0.779 ^b^	1.161 ^a^	0.05	0.001
C18:2n-6t	0.020 ^b,c^	0.042 ^a^	0.010 ^c^	0.033 ^a,b^	0.049 ^a^	0.00	0.000
C18:2n-6c	0.428 ^a,b^	0.459 ^a,b^	0.323 ^b,c^	0.281 ^c^	0.508 ^a^	0.02	0.002
C18:3n-6	0.011 ^a,b^	0.014 ^a^	0.006 ^b^	0.015 ^a^	0.012 ^a,b^	0.00	0.032
C18:3n-3	0.022	0.038	0.018	0.036	0.217	0.03	0.090
9c-11t CLA	0.023 ^b^	0.048 ^a^	0.022 ^b^	0.051 ^a^	0.039 ^a,b^	0.00	0.001
9c-11c CLA	0.008 ^b,c^	0.013 ^a^	0.004 ^c^	0.013 ^a,b^	0.011 ^a,b^	0.00	0.000
10t12c CLA	0.006 ^a,b^	0.009 ^a^	0.003 ^b^	0.009 ^a^	0.009 ^a^	0.00	0.001
C20:2	0.012 ^b^	0.019 ^a^	0.009 ^b^	0.013 ^a,b^	0.012 ^b^	0.00	0.000
C20:3n-6	0.030 ^c,d^	0.038 ^b,c^	0.018	0.072 ^a^	0.050 ^a,b^	0.00	0.000
C20:4n-6	0.071 ^b^	0.087 ^a,b^	0.076 ^a,b^	0.113 ^a,b^	0.116 ^a^	0.01	0.003
C20:3n-3	0.008 ^a^	0.004 ^b^	0.005 ^b^	0.008 ^a^	0.008 ^a^	0.00	0.000
C20:5n-3 (EPA)	0.010 ^c^	0.024 ^b^	0.022 ^a,b^	0.037 ^a^	0.026 ^a,b^	0.00	0.000
C22:2	0.008	0.014	0.008	0.010	0.012	0.00	0.377
C22:5n-3 (DPA)	0.043 ^b^	0.079 ^a^	0.034 ^b^	0.075 ^a^	0.081 ^a^	0.00	0.000
C22:6n-3 (DHA)	0.011 ^b^	0.022 ^a^	0.006 ^b^	0.015 ^a,b^	0.016 ^a^	0.00	0.000
n-6 PUFA	0.55 ^b^	0.63 ^a^	0.43 ^b^	0.50 ^b^	0.72 ^a^	0.03	0.001
n-3 PUFA	0.09 ^b^	0.17 ^a^	0.08 ^b^	0.17 ^a^	0.35 ^a^	0.03	0.027
PUFA/SFA	0.17 ^a^	0.17 ^a^	0.19 ^a^	0.13 ^b^	0.21 ^a^	0.01	0.012
n-6/n-3	6.09 ^a^	3.74 ^b^	5.23 ^a^	2.96 ^b^	2.27 ^b^	0.29	0.000
Desirable fatty acids	5.13 ^b^	7.31 ^a^	3.81 ^b^	7.91 ^a^	7.81 ^a^	0.39	0.002

SFA: saturated fatty acids; MUFA: monounsaturated fatty acids, PUFA: polyunsaturated fatty acids; Desirable fatty acids: MUFA + PUFA + C18:0; SE: Standard error. Different superscripts (^a–d^) indicate significant differences (*p* < 0.05) among breeds.

**Table 5 foods-14-03961-t005:** Total abundance of volatile compound families (values expressed as mean peak area × 10^3^) in the *Longissimus dorsi* muscle of five autochthonous cattle breeds from Southwest Spain.

	Breed					SE	*p*-Value
	Retinta	Pajuna	Marismeña	Berrenda en Colorado	Lidia		
Total carboxylic acid	394.12	358.34	347.52	363.34	334.30	37.12	0.987
Total alcohol	413.18	567.09	803.54	610.23	489.23	58.90	0.467
Total aldehydes	629.11	411.33	598.95	426.82	212.50	62.89	0.138
Total ketones	86.08 ^b^	69.55 ^b^	196.92 ^a^	88.55 ^b^	40.04 ^c^	11.07	0.001
Total sulfur compounds	40.08	21.20	22.22	25.08	19.97	3.41	0.185
Total furans	32.55	21.93	50.42	33.85	21.58	4.14	0.332
Total nitrogen compounds	167.74	97.19	218.08	195.89	60.54	19.20	0.052
Total pyrazines	70.58	41.68	123.00	44.44	27.65	10.60	0.117
Total pyrroles	20.66	15.23	17.81	14.31	7.49	1.84	0.104
Total esters	36.68 ^b^	48.03 ^b^	63.66 ^a,b^	97.63 ^a^	29.73 ^b^	5.55	0.003
Total aromatic hydrocarbons	29.86 ^b^	76.21 ^a^	96.76 ^a^	75.68 ^a,b^	33.67 ^b^	6.29	0.000
Total lactones	27.76	28.29	24.40	55.26	25.49	3.52	0.181

SE: Standard error. Different superscripts (^a–c^) indicate significant differences (*p* < 0.05) among breeds.

## Data Availability

The data presented in this study are openly available at https://idus.us.es/home.

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
