# Peer review of "Nutrient Composition of Autochthonous Beef from Southwest Spain"

_foods, 2025, doi:10.3390/foods14223961_

Round 1
Reviewer 1 Report
Comments and Suggestions for Authors
This study provides a comprehensive evaluation of the nutritional composition of beef from five autochthonous calving breeds in Southwest Spain (Retinta, Pajuna, Marismeña, Berrenda en Colorado, and Lidia). The analysis covers proximate composition, mineral content, fatty acid profiles, and volatile compounds. The findings highlight the favorable nutrient profiles and distinctive flavor traits of these Spanish native cattle breeds, emphasizing their value in sustainable production and conservation programs. This research effectively addresses a gap in the literature regarding the nutritional value of meat from native bovine breeds in Southwest Spain, contributing significantly to updating national food composition databases and supporting the preservation of genetic heritage. The article is well-structured, methods are detailed, results are clearly presented, and the discussion is insightful. However, there were some comments need to be addressed.
Specific comments:
- The sample size for Marismeña and Berrenda en Colorado breeds is only n=3. While the authors acknowledge the difficulty in obtaining samples due to the endangered status, such a small sample size may limit the robustness of statistical inferences and the generalizability of the findings. It would be beneficial to explicitly state this limitation more prominently in the discussion and suggest larger sample sizes for future research if feasible.
- Please adjust the position of Figure 1, and the legend of Figure 1 should be placed under the Figure 1.
- In Table 2, p-values are reported as "<0.001" or "ns". However, in Tables 3 and 4, p-values are generally given to three decimal places. It is advisable to maintain a consistent format for reporting p-values across all tables, for instance, uniformly using "< 0.001" for highly significant results.
- Line 294-295: The discussion attributes the highest fat content in Pajuna beef to "calves' castration, prolonged concentrate feeding, and an intermediate slaughter age". Providing more detailed mechanistic explanations of how these factors specifically influence fat deposition would strengthen this point.
- Line 440-444: Lidia beef has the lowest total fat content (2.98%) but significantly higher levels of SFA, MUFA, PUFA, and total FAMEs compared to other breeds. While the explanation that "FAMEs measure only esterified fatty acids, so fat content and FAME levels can be at variance depending on the lipid composition of the fat" is reasonable, a more detailed elaboration on the biological or biochemical reasons for this discrepancy, such as the relative proportion of non-esterified lipids (e.g., cholesterol) in Lidia beef fat, would be valuable.
Minor comments:
- Line 221-222: "The data are shown as mean values and the differences among the breeds studied were compared using one-way ANOVA followed by a post-hoc Tuckey test at p<0.05." could be streamlined to "Data were analyzed using one-way ANOVA followed by a post-hoc Tukey test (p<0.05) to compare differences among breeds."
- "The overall protein content in meat was within the range previously reported for Spanish breeds (21-24%), such as the Rubia Gallega (Table 2), Retinta and various crossbreeds found in Spanish cattle [12, 33]." could be rephrased for better flow, e.g., "The overall protein content in meat (21-24%) was consistent with previously reported ranges for Spanish breeds, including Rubia Gallega, Retinta, and various crossbreeds [12, 33]."
Author Response
ANSWERS REVIEWER 1 (Manuscript ID: foods-3974709)
Dear reviewer,
Thank you so much for your contribution. We have listened to your suggestions and we send the corrections to the document.
Best regards
Alberto Horcada
Specific comments:
- The sample size for Marismeña and Berrenda en Colorado breeds is only n=3. While the authors acknowledge the difficulty in obtaining samples due to the endangered status, such a small sample size may limit the robustness of statistical inferences and the generalizability of the findings. It would be beneficial to explicitly state this limitation more prominently in the discussion and suggest larger sample sizes for future research if feasible.
ANSWER: Reference to the small sample size of the Marismeña and Berrenda breeds in Colorado has been made in the Animals section (2.1) (see lines 102-104). Moreover, in the Conclusions section, a reference to the limited number of Marismeña and Berrenda breeds currently present in Spain and the need for research with larger sample sizes to obtain more robust results has been included. (see lines 578-579).
However, more studies about nutrient composition of some autochthonous beef from Southwest Spain is required because the information available is dramatically scarce and traditional production systems are heterogeneous and highly specific for each breed analyzed. (see lines 578-591).
- Please adjust the position of Figure 1, and the legend of Figure 1 should be placed under the Figure 1.
ANSWER: The legend of Figure 1 was placed under the Figure 1. (see line 111)
- In Table 2, p-values are reported as "<0.001" or "ns". However, in Tables 3 and 4, p-values are generally given to three decimal places. It is advisable to maintain a consistent format for reporting p-values across all tables, for instance, uniformly using "< 0.001" for highly significant results.
ANSWER: Changes to tables 2 and 3 have been made to the p-value (see changes in tables 2 and 3, the last columns)
- Line 294-295: The discussion attributes the highest fat content in Pajuna beef to "calves' castration, prolonged concentrate feeding, and an intermediate slaughter age". Providing more detailed mechanistic explanations of how these factors specifically influence fat deposition would strengthen this point.
ANSWER: An extensive literature describes that castration of young male calves has been found to enhance the accumulation of intramuscular fat in beef because castration technique decreases lipolysis in animal tissues while lipogenesis and lipid absorption increasing (see lines 324-327)
- Line 440-444: Lidia beef has the lowest total fat content (2.98%) but significantly higher levels of SFA, MUFA, PUFA, and total FAMEs compared to other breeds. While the explanation that "FAMEs measure only esterified fatty acids, so fat content and FAME levels can be at variance depending on the lipid composition of the fat" is reasonable, a more detailed elaboration on the biological or biochemical reasons for this discrepancy, such as the relative proportion of non-esterified lipids (e.g., cholesterol) in Lidia beef fat, would be valuable.
ANSWER: Interestingly, meat from the Lidia breed had the lowest total fat content (2.98%; Table 2), yet it exhibited significantly higher levels of SFA, MUFA, PUFA and total FAMEs compared with the other autochthonous breeds from southern Spain, second only to the Barrenda en Colorado breed (Table 4). These results suggest that the FAME content in the edible fat of the Lidia breed is higher than that of other local breeds, whereas other lipids, such as cholesterol, are lower. The total fat in meat includes all lipids, while FAMEs measure only esterified fatty acids, so fat content and FAME levels can be at variance depending on the lipid composition of the fat. Therefore, in leaner muscles such as those of the Lidia breed, which have a higher relative proportion of membrane phospholipids per unit tissue, compared to intramuscular fat, FAME yields can be comparatively higher even when the total NMR-detectable fat is lower. Conversely, fattier breeds accumulate larger triacylglycerol depots and higher proportions of non-esterifiable lipids (e.g., free cholesterol and sphingolipids), diluting the phospholipid fraction and resulting in lower FAMEs per 100 g tissue (see lines 475-481)
Minor comments:
- Line 221-222: "The data are shown as mean values and the differences among the breeds studied were compared using one-way ANOVA followed by a post-hoc Tuckey test at p<0.05." could be streamlined to "Data were analyzed using one-way ANOVA followed by a post-hoc Tukey test (p<0.05) to compare differences among breeds."
ANSWER: Data were analyzed using one-way ANOVA followed by a post-hoc Tukey test (p<0.05) to compare differences among breeds (see lines 238-239)
- "The overall protein content in meat was within the range previously reported for Spanish breeds (21-24%), such as the Rubia Gallega (Table 2), Retinta and various crossbreeds found in Spanish cattle [12, 33]." could be rephrased for better flow, e.g., "The overall protein content in meat (21-24%) was consistent with previously reported ranges for Spanish breeds, including Rubia Gallega, Retinta, and various crossbreeds [12, 33]."
ANSWER: The overall protein content in meat (21-24%) was consistent with previously reported ranges for Spanish breeds, including Rubia Gallega, Retinta, and various crossbreeds. (see lines 355-356)

Reviewer 2 Report
Comments and Suggestions for Authors
General Comments
This is a well-conducted and valuable study with minor areas for refinement. This manuscript should address the following points.
Specific Comments
Introduction
The background is informative and well-referenced. Consider slightly condensing the historical breed section to improve focus. and include at least one recent global comparison (e.g., compositional studies from South America or Africa) to emphasize the broader relevance of Spanish native breeds.
Materials and Methods
The methodology is well detailed. However, it would benefit from:
- Although welfare compliance is mentioned, clearly state the institutional ethics committee name, approval code, and date, in accordance with Foods author guidelines.
- Clarification on sample selection (randomization, age, and sex distribution justification): The manuscript would benefit from clearer justification of the limited sample size for endangered breeds. Please indicate how these small numbers still allow valid interpretation, perhaps by referencing population rarity or previous comparable studies.
- The manuscript should explicitly justify the inclusion of females (Lidia) and castrated males (Pajuna). Clarify how sex and physiological status may have influenced intramuscular fat and protein variability.
- Justification for small sample size in endangered breeds. Specify whether data were tested for normality and homogeneity of variance before ANOVA and PCA. Include brief details on how outliers were handled and report the proportion of total variance explained by discriminant functions.
- In the volatile compound section, cite a reference validating the chosen SPME conditions to strengthen methodological rigor.
- Provide a citation or brief explanation confirming that the selected SPME-GC-MS fiber type and conditions (time, temperature) are validated for beef volatile analysis to strengthen methodological reproducibility.
- Results and Discussion
- Results are extensive and well-interpreted, though a few sections (particularly the fatty acid discussion) could be slightly condensed.
- The discriminant analysis interpretation is insightful; however, consider summarizing the statistical meaning of “Function 1” and “Function 2” more concisely for general readers.
- Highlight key nutritional implications (e.g., low sodium, favorable n-6/n-3 ratio) earlier in the discussion for clarity.
- For trace minerals, it may help to include comparison with FAO or EFSA dietary reference intakes to contextualize nutritional significance.
- The Discussion is comprehensive but occasionally repetitive. Consider merging overlapping paragraphs (particularly in proximate and fatty acid results) to improve flow and emphasize key biological interpretations over descriptive text.
-
Strengthen interpretation of nutritional relevance: Expand on the public health significance of the findings-especially the “low sodium,” “high phosphorus,” and favorable “n-6/n-3” ratios-by comparing these values to EU or FAO nutritional reference limits.
- Add a short paragraph at the end of the Discussion summarizing the main limitations—small sample numbers for endangered breeds, unbalanced sex distribution, and possible environmental effects—to ensure transparency and realistic scope.
- Figures and Tables
- Tables are well formatted and legible. Ensure all abbreviations are defined in footnotes for accessibility.
- The discriminant analysis plot (Figure 2) could benefit from slightly larger axis labels and an explanation of the variance by each function, and briefly describe how the clustering supports breed differentiation in the text or caption.
- Language and Formatting
- The English language is clear and fluent. Minor editorial polishing (reducing repetition of “In this regard” or “in this context”) would enhance readability.
- Ensure consistent use of significant digits and spacing in all tables (e.g., “mg/100 g” instead of “mg /100g”).
- Conclusions
- Conclusions are justified and well-aligned with the data. However, the conclusion could be more impactful by linking the findings to future research, sustainable livestock practices, breed conservation programs, and consumer education regarding native beef nutritional value.
Author Response
ANSWERS REVIEWER 2 (Manuscript ID: foods-3974709)
Dear reviewer,
Thank you so much for your contribution. We have listened to your suggestions and we send the corrections to the document.
Best regards
Alberto Horcada
General Comments
This is a well-conducted and valuable study with minor areas for refinement. This manuscript should address the following points.
Specific Comments
- Introduction
The background is informative and well-referenced. Consider slightly condensing the historical breed section to improve focus. and include at least one recent global comparison (e.g., compositional studies from South America or Africa) to emphasize the broader relevance of Spanish native breeds.
ANSWER: Curiosity to know the composition of meat has developed widely in the world. For example, the quality and nutrient composition of “tropical beef” from cattle raised in the Tropics areas was reported by Rubio et al. (see lines 81-83)
- Materials and Methods
- Although welfare compliance is mentioned, clearly state the institutional ethics committee name, approval code, and date, in accordance with Foods author guidelines.
ANSWER: This study was conducted in several calve farms in Southwest Spain. All animal manipulations in this study were approved by the Ethics Committee of the University of Córdoba (Ref. CEIH-24-01). (see lines 128-130)
- Clarification on sample selection (randomization, age, and sex distribution justification): The manuscript would benefit from clearer justification of the limited sample size for endangered breeds. Please indicate how these small numbers still allow valid interpretation, perhaps by referencing population rarity or previous comparable studies.
ANSWER: Furthermore, since this document aims to report on the characteristics of the usual meat produced in Southwest Spain, the selection of animals was carried out randomly from different farms according to the production system for each breed under study, taking into account the sex and the animals' typical slaughter age (see lines 105-108)
- The manuscript should explicitly justify the inclusion of females (Lidia) and castrated males (Pajuna). Clarify how sex and physiological status may have influenced intramuscular fat and protein variability.
ANSWER: In the text, reference to inluence of sex on fat content in Castrated males of Pajuna breed was included:
An extensive literature describes that castration of young male calves has been found to enhance the accumulation of intramuscular fat in beef because castration technique decreases lipolysis in animal tissues while lipogenesis and lipid absorption increasing (see lines 324-327).
ANSWER: In reference to Lidia breed, reference to reduce fat content is reported aws following:
Although the Lidia breed animals were females, this breed showed the lowest IMF content (2.98%), a result consistent with an extensive production system characterized by high grazing utilization and low energy supplementation (see lines 332-334).
- Justification for small sample size in endangered breeds. Specify whether data were tested for normality and homogeneity of variance before ANOVA and PCA. Include brief details on how outliers were handled and report the proportion of total variance explained by discriminant functions.
Due to the endangered status of some breeds included in this study (Marismeña and Berrenda en Colorado), the small sample size was unavoidable. Both populations are in extreme danger of extinction, with very limited numbers of reproductive females, which makes sample collection highly restricted and logistically complex. For this reason (this limitation), all available representative animals were included to ensure biological and genetic representativeness within the constraints of conservation programs.
ANSWER: Before performing the ANOVA and PCA, data were tested for normality and homogeneity of variances using the Shapiro–Wilk and Levene’s tests, respectively. Outliers were examined using the interquartile range (IQR) method, defining potential outliers as values outside 1.5×IQR from the first or third quartile. No data were excluded, as all observations were within biologically plausible ranges. (see lines 238-244)
- In the volatile compound section, cite a reference validating the chosen SPME conditions to strengthen methodological rigor.
ANSWER: Reference was included in line 199: The volatile compounds of the meat were identified using the solid phase micro extraction (SPME) analysis technique [18].
[18] Gutiérrez-Peña, R.; García-Infante, M.; Delgado-Pertíñez, M.; Guzmán J.L.; Zarazaga L.A.; Simal S.; Horcada, A. Organoleptic and nutritional traits of lambs from Spanish mediterranean Islands raised under a traditional production system. Foods, 2022, 11(9), 1312. https://doi.org/10.3390/foods11091312
- Provide a citation or brief explanation confirming that the selected SPME-GC-MS fiber type and conditions (time, temperature) are validated for beef volatile analysis to strengthen methodological reproducibility.
ANSWER: The DVB/CAR/PDMS fiber has been reported as one of the most efficient coatings for the extraction of volatile compounds to cooked beef cuts, providing good reproducibility and a broad extraction range. Moreover, optimization studies recommend fiber coatings and extraction times in the range of 30–40 min for cooked meat matrices at 60 °C [19].
[19] Bueno, M.; Resconi, V.; Campo, M.M.; Ferreira, V.; Escudero, A. (2019). Development of a robust HS-SPME-GC-MS method for the analysis of solid food samples: Analysis of volatile compounds in fresh raw beef of differing lipid oxidation degrees. Food Chem., 2019, 281, 49–56. https://doi.org/10.1016/j.foodchem.2018.12.082
- Results and Discussion
- Results are extensive and well-interpreted, though a few sections (particularly the fatty acid discussion) could be slightly condensed.
ANSWER: The follow text “The results of this work provide information about individual profiles or different lipid ratios of cattle breeds from South of Spain. In fact, this information contributes to increasing the database of the lipid profile of meat from several European cattle breeds reported by Moreno et al. [XX] and Campo et al. [XX] reared under traditional management systems” ” was eliminated.
- The discriminant analysis interpretation is insightful; however, consider summarizing the statistical meaning of “Function 1” and “Function 2” more concisely for general readers.
ANSWER: More information about interpretation of discriminant analysis was included.
Function 1 (vertical axis) explain 55.8%of the variability among breeds differentiating the breeds according to carcass weight, protein and ash content, several mineral concentrations (particularly K, Na, Zn, and P), and several polyunsaturated fatty acids (PUFAs), including C18:3n-6, C20:3n-3, and C18:2n-6. This horizontal separation clearly discriminated between breeds not at risk of extinction (Retinta and Lidia), located on the right side of the plot, and endangered breeds (Berrenda en Colorado, Pajuna, and Marismeña), positioned on the left. Function 2 (horizontal axis) explain 25.2% of variability among studied breeds. (see lines 260-267)
It is remarkable that function 1 and function 2 include carcass weight, protein and several polyunsaturated fatty acids in meat to explain variability among calves breeds from Southwest Spain (see lines 279-281)
- Highlight key nutritional implications (e.g., low sodium, favorable n-6/n-3 ratio) earlier in the discussion for clarity.
ANSWER: Reference to sodium index and n-6/n-3 were included near discussion site.
According to regulations on nutrition claims established by the European Commission [31], intake of “low in sodium” (<120 mg/100 g) content in meat for prevent cardiovascular health diseases is proposed. (see lines 406-408).
…….. desirable fatty acids (EPA, DPA, DHA and CLA)….. (see lines 485-486)
In order to prevent cardiovascular diseases in human, several studies have suggested that a maximum n-6/n-3 PUFA ratio of 4 is recommended for human health benefits [52] (see lines 493-494).
- For trace minerals, it may help to include comparison with FAO or EFSA dietary reference intakes to contextualize nutritional significance.
ANSWER: In order to guarantee health of consumers, the European Food Safety Authority (EFSA) [43] reported the dietary reference intakes level of trace mineral in meat as following: daily Ca intake <2,500mg/d and Na < 2,000mg/d; P around 550mg/d; K around 3,500mg/d; Mg around 250mg/d; Zn < 25mg/d; Se around 255µg/d and Fe between 6-13 mg/d. (see lines 390-394).
- The Discussion is comprehensive but occasionally repetitive. Consider merging overlapping paragraphs (particularly in proximate and fatty acid results) to improve flow and emphasize key biological interpretations over descriptive text.
ANSWER: To a better comprehensive results, in discussion section changes have been made. Moreover, biological interpretations on the nutritional value of meat have been reported (See paragraphs in red in sections 3.2 and 3.4 mainly).
- Strengthen interpretation of nutritional relevance:Expand on the public health significance of the findings-especially the “low sodium,” “high phosphorus,” and favorable “n-6/n-3” ratios-by comparing these values to EU or FAO nutritional reference limits.
ANSWER: Reference to low intake sodium on human heath is reported in lines 406- 408 as following: According to regulations on nutrition claims established by the European Commission, intake of “low in sodium” (<120 mg/100 g) content in meat for prevent cardiovascular health diseases is proposed.
ANSWER: Reference to low intake phosphorus on human heath is reported in lines 416- 418 as following: Phosphorus concentrations in beef from Southwest Spanish autochthonous breeds, which ranged between 220–235 mg/100 g, mean that the product is considered a “high phosphorus content” food.
ANSWER: Reference of the favorable n-6/n-3 index is detailed in lines 493-500 as following: In order to prevent cardiovascular diseases in human, several studies have suggested that a maximum n-6/n-3 PUFA ratio of 4 is recommended for human health benefits. In this context, meat from the Pajuna, Berrenda en Colorado, and Lidia breeds exhibited a more favorable n-6/n-3 ratio than Retinta and Marismeña breeds. Specifically, the first three breeds showed ratios below 4, considered beneficial, whereas Retinta and Marismeña had higher ratios ranging from 5.23 to 6.09 (Table 4).
- Add a short paragraph at the end of the Discussion summarizing the main limitations—small sample numbers for endangered breeds, unbalanced sex distribution, and possible environmental effects—to ensure transparency and realistic scope.
ANSWER: In conclusion section a short paragraph summarizing the main limitarions of study has been included. However, more studies about nutrient composition of some autochthonous beef from Southwest Spain is required because the information available is dramatically scarce and traditional production systems are heterogeneous and highly specific for each breed analyzed (see lines 578-581).
- Figures and Tables
- Tables are well formatted and legible. Ensure all abbreviations are defined in footnotes for accessibility.
ANSWER: All abbreviations are defined in footnotes.
- The discriminant analysis plot (Figure 2) could benefit from slightly larger axis labels and an explanation of the variance by each function, and briefly describe how the clustering supports breed differentiation in the text or caption.
ANSWER: Discriminant labels are larger. Variance explanation values have been incorporated into each function (see Figure 2)
- Language and Formatting
- The English language is clear and fluent. Minor editorial polishing (reducing repetition of “In this regard” or “in this context”) would enhance readability.
ANSWER: The English language has been reviewed and expressions as "In this regard” or “in this context” were deleted.
- Ensure consistent use of significant digits and spacing in all tables (e.g., “mg/100 g” instead of “mg /100g”).
ANSWER: Significant digits and spacing in all tables and text has been reviewed.
- Conclusions
Conclusions are justified and well-aligned with the data. However, the conclusion could be more impactful by linking the findings to future research, sustainable livestock practices, breed conservation programs, and consumer education regarding native beef nutritional value.
ANSWER: References to future research and the application of findings on meat labeling or plans conservation for native breeds have been included in the conclusions. (see lines 579-586).
